**Subject Category:**
Biology (whole organism)

palaeontology

exceptional preservation, Herefordshire Lagerstätte, Porifera, Reticulosa, Silurian

**Author for correspondence:**
Mark D. Sutton
e-mail: m.sutton@imperial.ac.uk

# Three-dimensionally preserved soft tissues and calcareous hexactins in a Silurian sponge: implications for early sponge evolution

Ardianty Nadhira[1], Mark D. Sutton[1],
Joseph P. Botting[2,3], Lucy A. Muir[3], Pierre Gueriau[4,5],
Andrew King[6], Derek E. G. Briggs[7], David J. Siveter[8]
and Derek J. Siveter[9,10]

[1]Department of Earth Sciences and Engineering, Imperial College London, London SW7 2BP, UK
[2]Nanjing Institute of Geology and Palaeontology, 39 East Beijing Road, Nanjing 210008, People's Republic of China
[3]Department of Natural Sciences, Amgueddfa Cymru—National Museum Wales, Cathays Park, Cardiff CF10 3NP, UK
[4]IPANEMA, CNRS, Ministère de la Culture, UVSQ, USR 3461, Université Paris-Saclay, 91192 Gif-sur-Yvette, France
[5]Institute of Earth Sciences, University of Lausanne, Géopolis, CH-1015 Lausanne, Switzerland
[6]SOLEIL synchrotron, 91192 Gif-sur-Yvette, France
[7]Department of Geology & Geophysics, and Yale Peabody Museum of Natural History, Yale University, PO Box 208109, New Haven, CT 06520-8109, USA
[8]School of Geography, Geology and the Environment, University of Leicester, Leicester LE1 7RH, UK
[9]Earth Collections, University Museum of Natural History, Oxford OX1 3PW, UK
[10]Department of Earth Sciences, University of Oxford, South Parks Road, Oxford OX1 3AN, UK

MDS, 0000-0002-7137-7572; PG, 0000-0002-7529-3456;
DEGB, 0000-0003-0649-6417; DaJS, 0000-0002-1716-5819;
DeJS, 0000-0002-5305-2192

Sponges (Porifera), as one of the earliest-branching animal phyla, are crucial for understanding early metazoan phylogeny. Recent studies of Lower Palaeozoic sponges have revealed a variety of character states and combinations unknown in extant taxa, challenging our views of early sponge morphology. The Herefordshire Konservat–Lagerstätte yields an abundant, diverse sponge fauna with three-dimensional preservation of spicules and soft tissue. *Carduispongia pedicula* gen. et sp. nov. possesses a single layer of hexactine spicules arranged in a regular orthogonal

network. This spicule type and arrangement is characteristic of the reticulosans, which have traditionally been interpreted as early members of the extant siliceous Class Hexactinellida. However, the unusual preservation of the spicules of *C. pedicula* reveals an originally calcareous composition, which would be diagnostic of the living Class Calcarea. The soft tissue architecture closely resembles the complex sylleibid or leuconid structure seen in some modern calcareans and homoscleromorphs. This combination of features strongly supports a skeletal continuum between primitive calcareans and hexactinellid siliceans, indicating that the last common ancestor of Porifera was a spiculate, solitary, vasiform animal with a thin skeletal wall.

## 1. Introduction

The origin of the sponges (Porifera) is inextricably bound up with the origin of the Metazoa. They are traditionally considered to be the earliest-branching metazoans, but an alternative hypothesis holds that they may be secondarily simplified from organisms of ctenophore- or cnidarian-grade [1]. The results of analyses of the skeletal morphology and genetics of living sponges [2–4] are inconclusive on this question.

There are clear distinctions between the skeletal morphology of the four extant sponge classes [5]. Demospongiae possess siliceous spicules, characteristically tetractonid/monaxonid with a triangular or hexagonal axial filament/canal, and additional microscleres or an organic framework skeleton. Hexactinellida have hexactine siliceous spicules with a square axial filament/canal and additional, distinct microscleres. Calcarea possess calcite spicules with no axial filament, and the related Homoscleromorpha possess siliceous tetractine/monaxon spicules (where present) with a diffuse axial region rather than a discrete filament. Siliceous spicules are secreted onto the axial filament, whereas calcareous spicules are secreted inside an organic sheath surrounding the spicule (see Uriz [6] for a review). Intermediates between these states are unknown in extant taxa; without fossil evidence their evolutionary pathways would be unresolved, including the question of whether spicules are homologous between classes.

Sponges feed with choanocyte cells, which form a cell layer (choanoderm) that normally lines an organized array of chambers. The aquiferous systems of sponges (i.e. how the choanoderm and the water flow over it are organized) are traditionally classified into three grades [7]. These are ascon (the simplest, in which a single choanoderm layer lines the atrium), sycon (with folding of choanoderm to increase surface area) and leucon (the most complex, in which choanoderm lines a network of small choanocyte chambers). This scheme is now widely regarded as inadequate to capture the diversity of structures and some authors (e.g. [8]) recognize additional categories, including a sylleibid grade, which is leuconoid-like but with choanocyte chambers clustered around primary exhalent conduits. Among siliceous sponges, the simplest structures are those of thin-walled hexactinellids such as *Farrea* [9], in which the skeletal meshwork defines an array of rounded, simple chambers lined by choanocytes – effectively a syconid condition, but without folding of the wall. The sylleibid architecture is similar, but with the addition of small sub-chambers lining the margin of these chambers and feeding into them; it approaches a leuconoid level of complexity. Despite its importance in sponge biology, however, it has only been possible to reconstruct the aquiferous system in fossil sponges which have close modern relatives as a basis for comparison.

Data from exceptionally preserved fossils are required to resolve the evolution of spicules and the aquiferous system. Articulated sponge fossils preserving soft tissues, such as those in the Burgess Shale and Chengjiang biotas (e.g. [10,11]), typically lack fine detail and internal morphology. The data available, however, provide clear evidence of phylogenetic complexity and extinct character combinations. These include examples of bimineralic spicules combining silica and calcite [12]; a species with spicules that are diagnostic of two distinct classes [13]; a body architecture characteristic of Demospongiae associated with hexactine spicules [14]; and well-developed body symmetry absent in modern taxa [15]. A recent assessment of the contribution of fossils to sponge phylogeny and the likely position of groups of fossil sponges reconstructed the last common ancestor of Porifera as a thin-walled sponge with bimineralic hexactine spicules and a relatively simple aquiferous system (ascon–sycon) [16].

The Silurian Herefordshire Lagerstätte [17] preserves a diverse fauna of invertebrates in three dimensions, in most cases with soft tissues. The fossils are studied through virtual palaeontology techniques [18]. Sponges are well-represented by at least 20 species, which comprise 15–20% of specimens. The most abundant of these species is established herein as *Carduispongia pedicula* gen.

et sp. nov. This taxon provides three-dimensional documentation of a novel combination: a relatively complex aquiferous system and a spicule network of calcareous hexactins.

# 2. Material and methods

The fossils of the Herefordshire Konservat–Lagerstätte occur in calcareous nodules in a volcaniclastic deposit [17]. They are preserved as calcitic infills in three dimensions, but exhibit little or no X-ray contrast between fossil and matrix and cannot be imaged with conventional X-ray Microtomography. Certain taxa, however, are proving amenable to phase-contrast Synchrotron X-ray Microtomography [19]. The holotype of *C. pedicula* (Oxford University Museum of Natural History specimen OUMNH C.36010) was scanned with the PSICHÉ beamline of the SOLEIL synchrotron (Saint-Aubin, France) using a pink beam (63–69 keV), a propagation distance of 200 mm, and 6000 projections. The limited field of view available (12 mm × 3.6 mm) was extended horizontally by positioning the rotation axis off-centre, and extended vertically by recording a series of acquisitions with vertical movement of the sample. The volume (6.5 µm voxel size) was reconstructed from the combined radiographs using PyHST2 software [19], with a Paganin phase retrieval algorithm [20]. Internal features are difficult to discern in the virtual tomographic data. Data from the holotype were supplemented by physical-optical tomography of OUMNH C.36032 to elucidate internal structures not resolved by synchrotron scanning, using the method described by Sutton *et al.* [18] at 30 µm grind intervals. Preservation of internal structures varies throughout the body, in part due to incomplete filling of internal spaces by sediment. One relatively well-preserved section has been reconstructed in detail (figure 1*g–i*). Segmentation and 3D rendering of 'virtual fossils' were performed for both datasets using the SPIERS software suite [21], and an isolated hexactin spicule (figure 3) was rendered and slightly smoothed using 3D Slicer (https://www.slicer.org/). Two-dimensional photographs were obtained with a Leica DFC420 digital camera mounted on a Leica MZ8 binocular microscope; specimens were immersed in a thin layer of water to enhance contrast, and digital post-processing of contrast was applied using GIMP (http://www.gimp.org). Elemental mapping of OUMNH C.36061 was performed using a Zeiss EVO 15LS scanning electron microscope/energy dispersive X-ray analysis (SEM/EDX) system at the Natural History Museum, London.

# 3. Systematic palaeontology

Phylum Porifera [22]
Class Calcarea?
'Order Reticulosa' [23]
Genus *Carduispongia* gen. nov.
Etymology: Latin, *Carduus* (thistle) + *spongia* (sponge), referring to the thistle-like appearance. Gender feminine.

**Diagnosis**

Thin-walled, ovoid body with a single spicule layer consisting of a regular, orthogonal reticulate network of calcareous hexactine spicules that decrease in size towards base and osculum. Some spicules bear elongate prostalial rays that curve upwards, and long gastral rays. Apical region with numerous shorter distal rays, not limited to the oscular margin.

**Remarks**

Reticulosa was originally established as an extinct lineage of hexactinellids [23], but it has become a 'wastebasket taxon' for thin-walled sponges with a semi-regular array of hexactin-based spicules, and is undoubtedly para- and/or polyphyletic as currently composed [16]. A revision of the taxonomy of Palaeozoic sponges is beyond the scope of this study, and we place *Carduispongia* informally within the Reticulosa for consistency with previous work. This provisional placement does not imply a close phylogenetic relationship to Hexactinellida.
Type species. *Carduispongia pedicula* sp. nov., by monotypy.

*Carduispongia pedicula* sp. nov., figures 1, 2–6*f–i*

**Diagnosis**

As for genus.

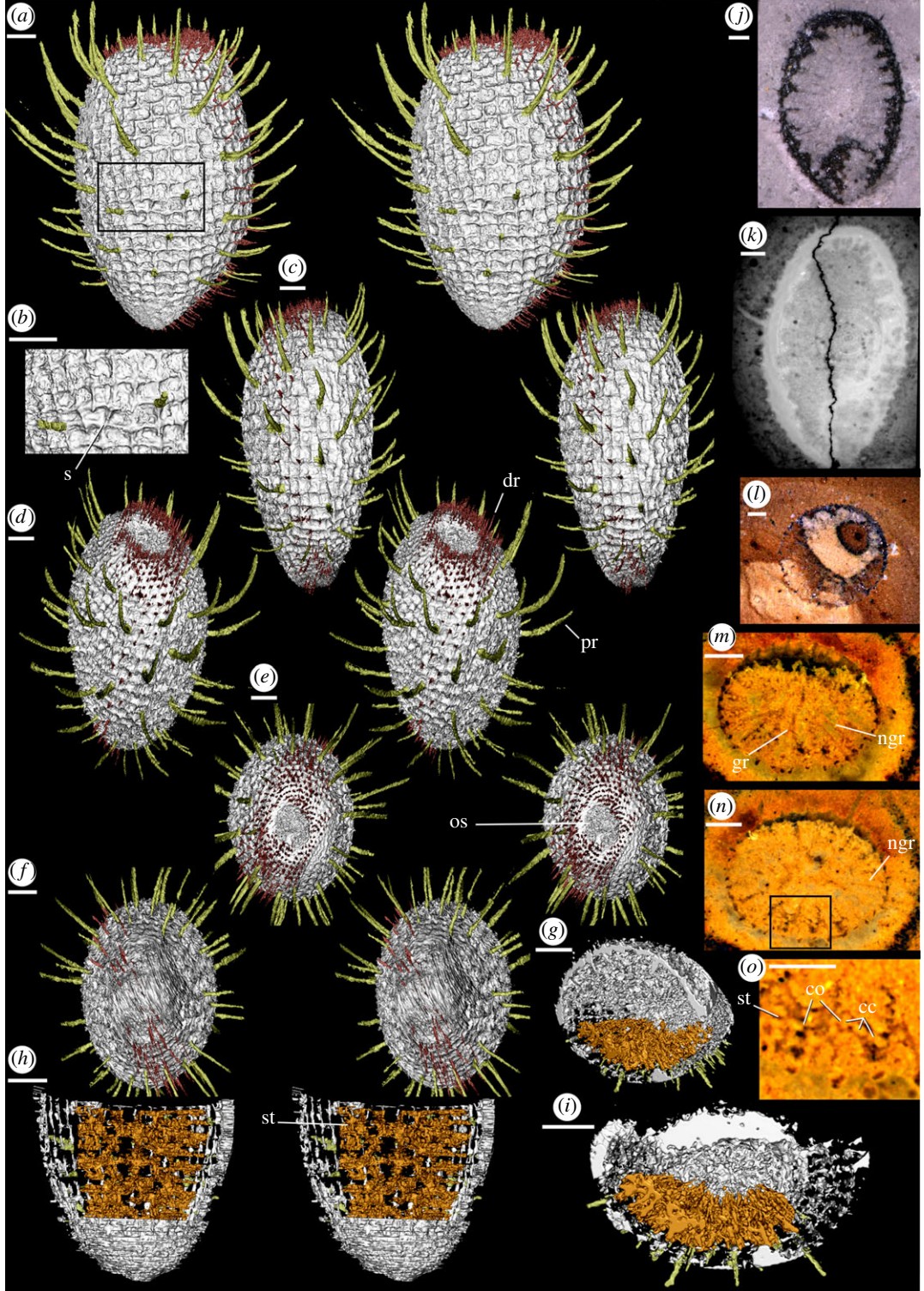

**Figure 1.** *Carduispongia pedicula* gen. et sp. nov., virtual reconstructions (*a–i*), tomograms (*k,m–o*) and photographs of rock surfaces (*j,l*). (*a–f, j,k*) OUMNH C.36010, complete specimen, holotype; (*a*) virtual stereo-pair of broadest lateral aspect, (*b*) magnified view of spicule network, (*c*) stereo-pair of narrowest lateral aspect, (*d*) oblique stereo-pair, osculum-upwards, (*e*) osculum stereo-pair, (*f*) basal section stereo-pair, (*j*) longitudinal cross-section, (*k*) slice from synchrotron dataset with first-order distal ray visible (left). (*l*) OUMNH C.36078, near-circular transverse section of specimen with aspect ratio of 1.06. (*g–i,m–o*) OUMNH C.36032, half-complete specimen with transverse views of internal details; (*g*) transverse-oblique view, (*h*) vertical stereo-pair with view of root, section of dermal layer removed to expose soft tissue, (*i*) transverse view, section of dermal layer removed, (*m,n*) physical-optical tomograms showing transverse sections with internal details, particularly gastral rays, (*o*) view of soft tissues and choanocyte chambers. All scale bars 1 mm. cc, choanocyte chambers; co, inter-chamber openings; dr, (non-hypertrophied) distal ray; gr, hypertrophied gastral ray; ngr, non-hypertrophied gastral ray; os, osculum; pr, prostalia (hypertrophied distal rays); s, spicule; st, soft tissue.

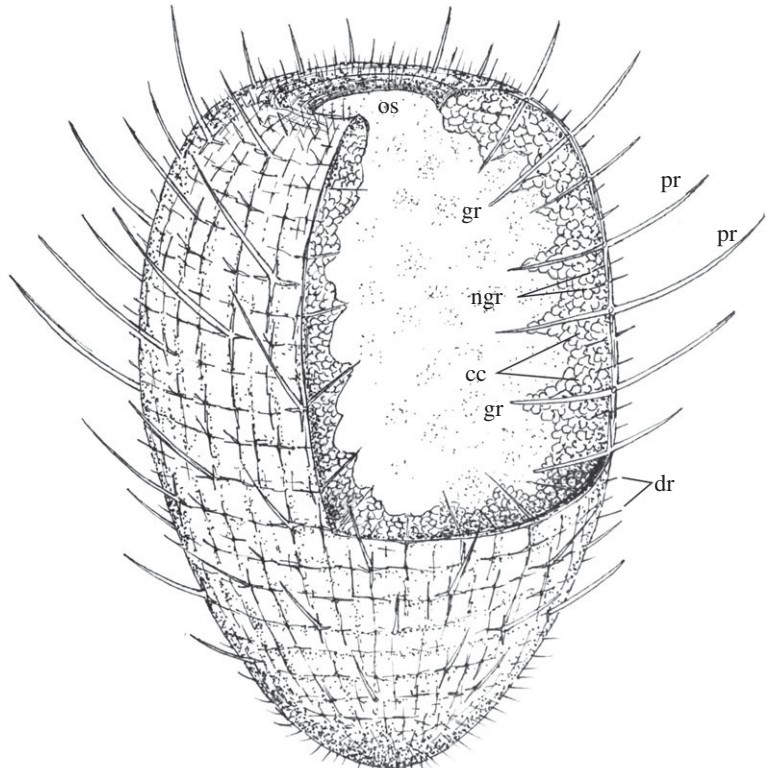

**Figure 2.** *Carduispongia pedicula* gen. et sp. nov., schematic reconstruction showing two vertical sections through the osculum to near the midpoint, and a transverse section intersecting this. Abbreviations as figure 1. Not to scale.

### Etymology

From Latin *pediculus* (louse), alluding to the fancied resemblance of the first specimen discovered (OUMNH C.36075, figure 5*g*).

### Material

Holotype: OUMNH C.36010, figure 1*a–f,j,k*. Complete specimen, length 10.7 mm, split to reveal longitudinal cross-section showing internal structures including gastral rays, and reconstructed using synchrotron phase-contrast scanning.

Paratypes: OUMNH C.36032 (reconstructed through physical-optical tomography), figure 1*g–i, m–o*. OUMNH C.36060, figure 6*i*. OUMNH C.36074, figure 6*f*. OUMNH C.36075, figure 6*g*. OUMNH C.36078, figure 1*l*. OUMNH C.36081, figure 6*h*.

Other material: approximately 133 additional specimens in the Oxford University Museum of Natural History.

### Locality and horizon

All specimens from the Coalbrookdale Formation, Wenlock Series (*ca* 430 Ma), Silurian System, Herefordshire, UK.

### Description

Body shape and size are difficult to determine for non-reconstructed specimens, but all observed sections are compatible with the ovoid body form of the holotype (figure 1*a–f,j,k*). The largest specimens are probably 13–14 mm in maximum height. The maximum height of the holotype is 10.7 mm excluding the spines of the osculum- and root-tufts, and the maximum width is 6.5 mm at 5.6 mm from the root (figure 1*a*). The maximum width of OUMNH C.36032 (figure 1*g–i, m–o*) is not reconstructed, but a transverse section 5.4 mm from the root has a width of 6.5 mm. Both reconstructed specimens are somewhat flattened laterally, aspect ratios in transverse section are 1.17 and 1.25 in the holotype and OUMNH C.36032 respectively. This flattening is likely *in vivo* as it is parallel to the body axis, but taphonomic flattening cannot be excluded. Determination of precise aspect ratios in non-reconstructed specimens is not possible without knowledge of the angle of section, but some sections have aspect ratios below 1.17 (e.g. OUMNH C.36078, figure 1*l*, aspect ratio 1.06) and these specimens may be less flattened than the holotype.

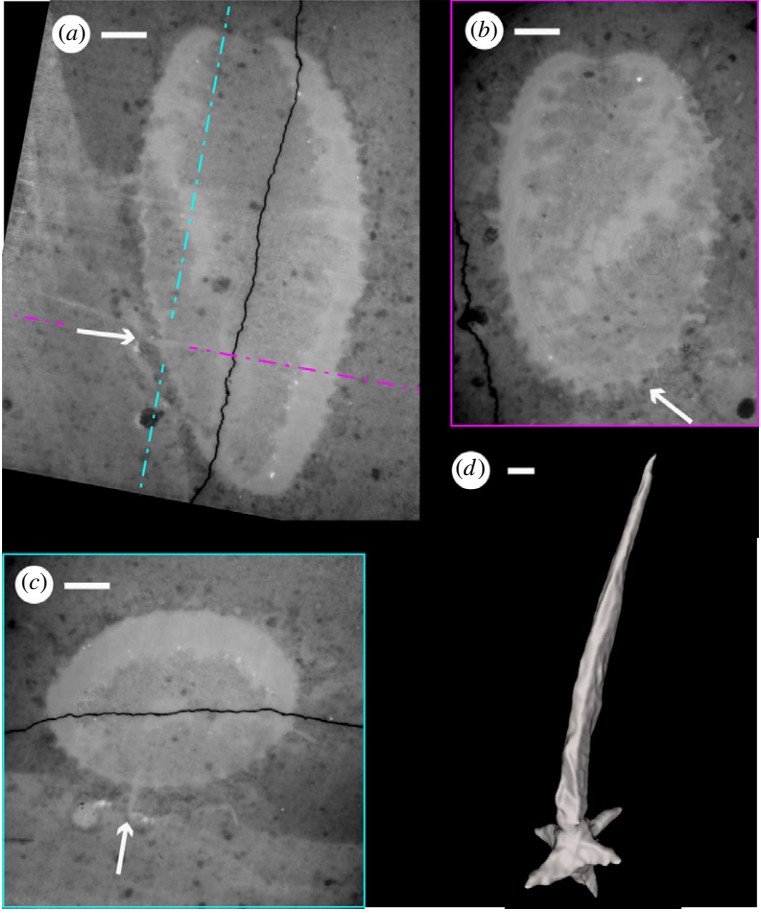

**Figure 3.** Single spicule of *Carduispongia pedicula* gen. et sp. nov., OUMNH C.36010 (holotype). (*a–c*) Tomograms, with spicule indicated by white arrow, scale bars 1 mm. (*a*) Tomogram showing positions of section of (*b,c*). (*b*) Resliced tomogram along blue dashed line of (*a*), orthogonal to (*a,c*). (*c*) Resliced tomogram along purple dashed line of (*a*), orthogonal to (*a,b*). (*d*) Virtual reconstruction of single spicule (arrow in (*a–c*) showing hexactin form (distal portions of gastral and dermal rays not recovered); scale bar 25 µm.

The apex of the body in the holotype measures 3.0 mm by 2.5 mm. A sunken central subcircular lacuna, of maximum diameter 1.4 mm and depth 0.3 mm, is interpreted as the osculum (figures 1*e* and 2). The bottom of the body tapers gradually to a rounded, obtuse termination (figures 1*f* and 2).

The preserved spicule network comprises a single layer, probably hypodermal in position. Quadrate reticulation is clear on the dermal surface of the holotype (figures 1*a–c*, 2 and 4). Spicules are hexactins (figure 3) without clear size orders; they are preserved as calcite, and are inferred to have been purely calcareous in original mineralogy (see discussion). Spicule count increases with sponge diameter. The holotype possesses 32 spicules around the body in transverse section at 3 mm from the base, 40 spicules at 5.6 mm from the base (the point of greatest diameter), and 37 spicules at 9 mm from the base. Tracing 'columns' of spicules vertically, identifiable points of disturbance manifest as trifurcations of columns (figure 4*d,e*). Spicule intercalation to accommodate increased spicule numbers towards the centre of the sponge thus occurs by the introduction of a new pair of spicule columns rather than by a single extra column. Angles between spicule rays are 90° except for rare variations in ray angles of up to 40°; these occur at points of column intercalation (figure 4*d,e*). Over most of the body of the holotype, spicule spacing is approximately 500 µm and the lateral rays have a typical basal diameter of approximately 90 µm. Within 1 mm of both the osculum and the base, the spicule network becomes increasingly tightly spaced with a decrease in spicule spacing to 200 µm or less and a fining in lateral-ray basal-diameter to approximately 50 µm, and spicules become smaller. The terminations of the lateral rays (i.e. those parallel to the dermal surface) coincide with those of adjacent spicules at junctions where the ends of the lateral rays converge or overlap (figure 4*f*). Nodes in the spicule network hence alternate between spicule bosses (bearing prostalia or shorter distal rays; green/red circles in figure 4*c–e*) and 'overlap junctions' where ray ends overlap (white circle in figure 4*c–e*). Overlap junctions are superficially boss-like, but their nature is apparent in well-preserved sections of

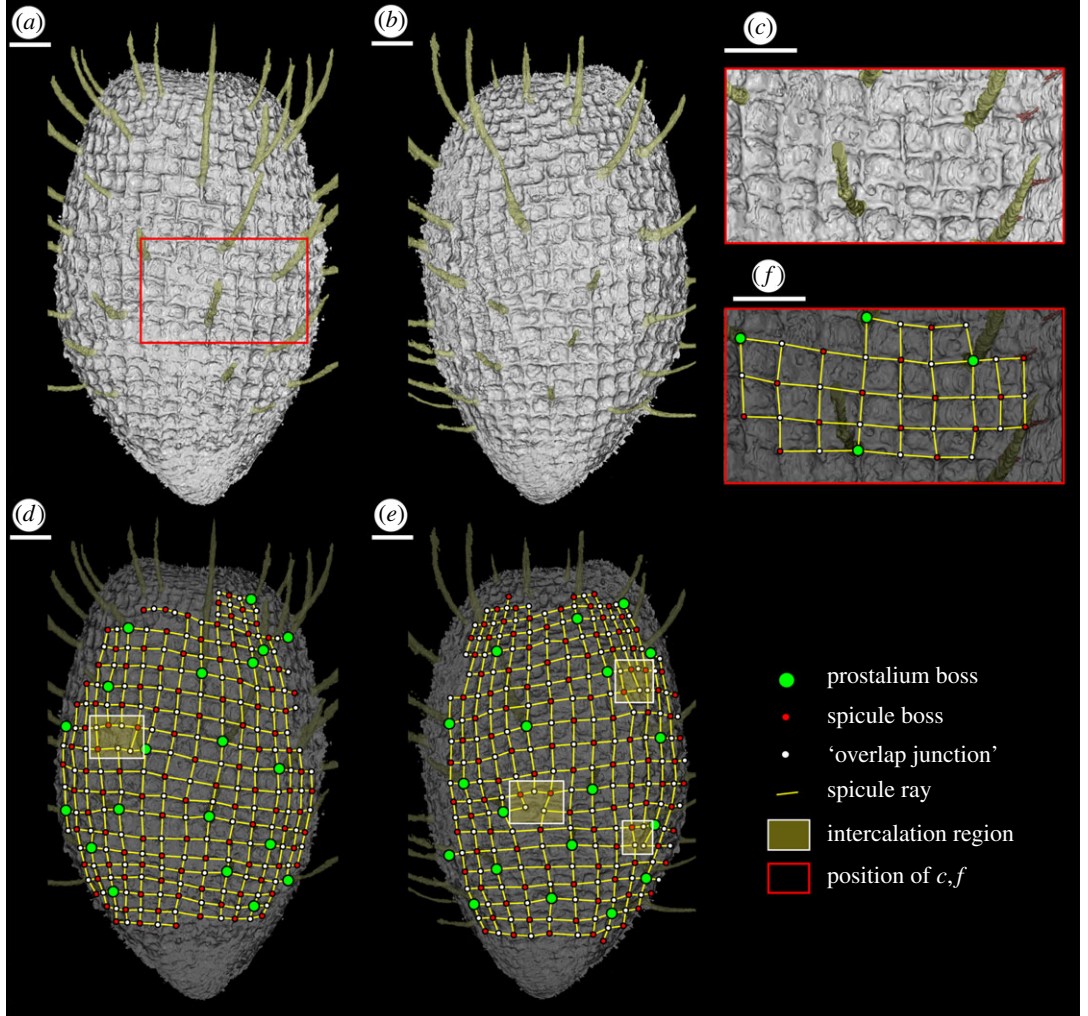

**Figure 4.** *Carduispongia pedicula* gen. et sp. nov. OUMNH C.36010 (holotype). Virtual reconstructions of complete specimen, (*d–f*) with interpretations of spicule network overlain. (*a,b*) Lateral views at 180° to each other, with non-hypertrophied distal rays removed. (*c*) Detail of well-preserved portion of spicule network showing ray-overlap. (*d–f*) Interpretations of (*a–c*), showing positions of prostalium bosses, non-prostalial bosses, 'overlap junctions' (false bosses where distal tips of rays overlap) and regions of column intercalation. All scale bars 1 mm.

the network (figure 4*c,f*). The spicule arrangement is thus quincunxial. At overlap junctions, the ray extending towards the root from the boss above is typically more prominent than that extending towards the osculum from the boss below (figure 4*c*), implying that the former overlies the latter.

Distal rays occur in two size categories (figures 1*a–f*, *k* and 2). Elongate prostalia are up to 3.3 mm long and 200 µm in diameter, typically curving upwards by 10–40° from the dermal perpendicular, and gradually tapering to a point. They are absent within 2 mm of the base and 0.5 mm of the top of the holotype (figure 1*e*), but occur on approximately 20% of the remaining spicules. The prostalia are sub-evenly spaced (figures 1*d*, 2 and 4) without a clear distribution pattern. Smaller non-hypertrophied distal rays, typically approximately 600 µm long and 50 µm in diameter, are apparently borne by all the other spicules, although they are not always preserved. In the basal region of the holotype, individual spicules are not visible in the dermal surface (figure 1*f*), although non-hypertrophied distal rays reveal their presence. Dermal spicules are visible at the top of the specimen (figure 1*e*); their cryptic nature at the base may reflect relatively deep embedding within soft tissues.

Gastral rays of spicules protrude into the central cavity and are most clearly visible in section (figure 1*m,n*). The longest gastral rays, which arise from spicules bearing prostalia, are up to 1.8 mm long and approximately 100 µm in diameter. Unlike prostalia, these gastral rays do not curve. Smaller gastral rays are approximately 50 µm in diameter, and occur on spicules bearing non-hypertrophied distal rays; they are inconsistently preserved. We infer that all spicules were hexactins bearing both distal and gastral rays (figure 2).

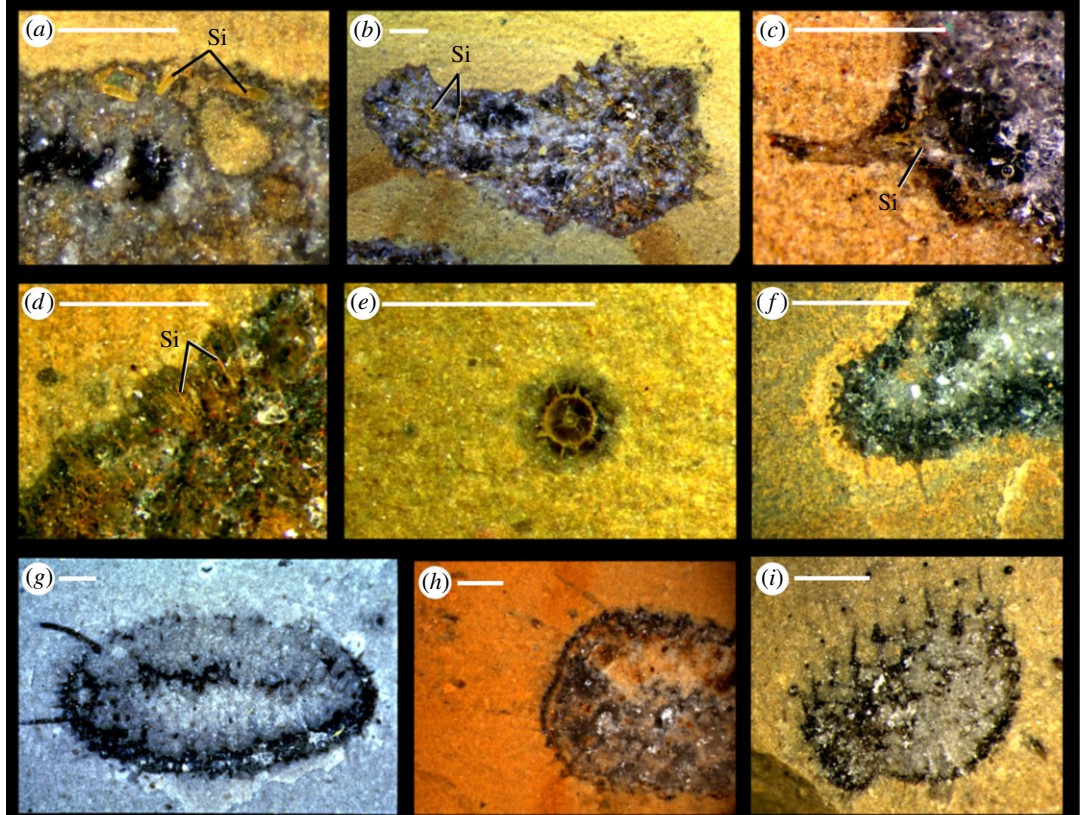

**Figure 5.** Photographs of rock surfaces displaying preservation of siliceous (*a–e*) and calcareous (*f–i*) specimens in the Herefordshire Lagerstätte. (*a–d*) Siliceous sponges with silica preserved as yellow ankerite; (*a*) OUMNH C.36080, unidentified species 1. (*b*) OUMNH C.36002, unidentified species 2. (*c*) OUMNH C.36079, unidentified species 2. (*d*) OUMNH C.36077, unidentified species 3. (*e*) OUMNH C.36076, radiolarian with test preserved as yellow ankerite. (*f–i*) *Carduispongia pedicula* gen. et sp. nov. (*f*) OUMNH C.36074. (*g*) OUMNH C.36075. (*h*) OUMNH C.36081. (*i*) OUMNH C.36060. Si; siliceous spicules, preserved discretely inside calcite fossil mass. All scale bars 1 mm.

The spicules are preserved as calcite; this is evident through optical inspection in all specimens examined. Supporting evidence is provided by elemental maps of two specimens (figure 5) which reveal a concentration of calcium and a depletion in iron and silicon in the spicules; this combination is compatible with a calcite mineralogy but not with either silica or ankerite (see discussion below). No trace of any axial structure was observed within the spicules either optically or in elemental maps.

Impersistently preserved soft tissues adhere to the gastral rays, forming a poorly defined 'platform' of soft tissue reconstructed as extending inwards for approximately 1 mm, or 25% of the maximum width of the transverse section (figures 1*g–i* and 2). This material follows the spicule network loosely, thinning away from gastral rays, and exhibiting quadrate gaps reflecting spicule reticulation. Where preservation is good, clusters of sub-spherical sediment-filled structures are evident within this tissue (figure 1*o*; mean diameter 82 μm, see electronic supplementary material, table S1); many are joined by gaps in their walls. The distribution, morphology and size of these structures implies that they represent large choanocyte chambers which were filled by sediment during burial.

## 4. Discussion

Siliceous biomineralized elements occur in other taxa from the Herefordshire deposit, including radiolarians ([24]; figure 6*e*) and numerous as-yet-unstudied sponges (figure 6*a–d*). These elements are preserved in a distinctive style, replaced or partially replaced by the carbonate mineral ankerite (Ca(Fe,Mg,Mn)(CO$_3$)$_2$), which is yellow in colour ([25], see also [24]). Siliceous elements, either in ankerite or as primary silica, form sharply bounded structures which are clearly distinct from areas of calcite within Herefordshire fossils (figure 6*a–d*). The spicules in *Carduispongia* lack ankerite or primary silica. They are preserved purely as calcite (see description), and do not form discrete elements differentiated from the rest of the fossil, suggesting that recrystallization of the spicules and

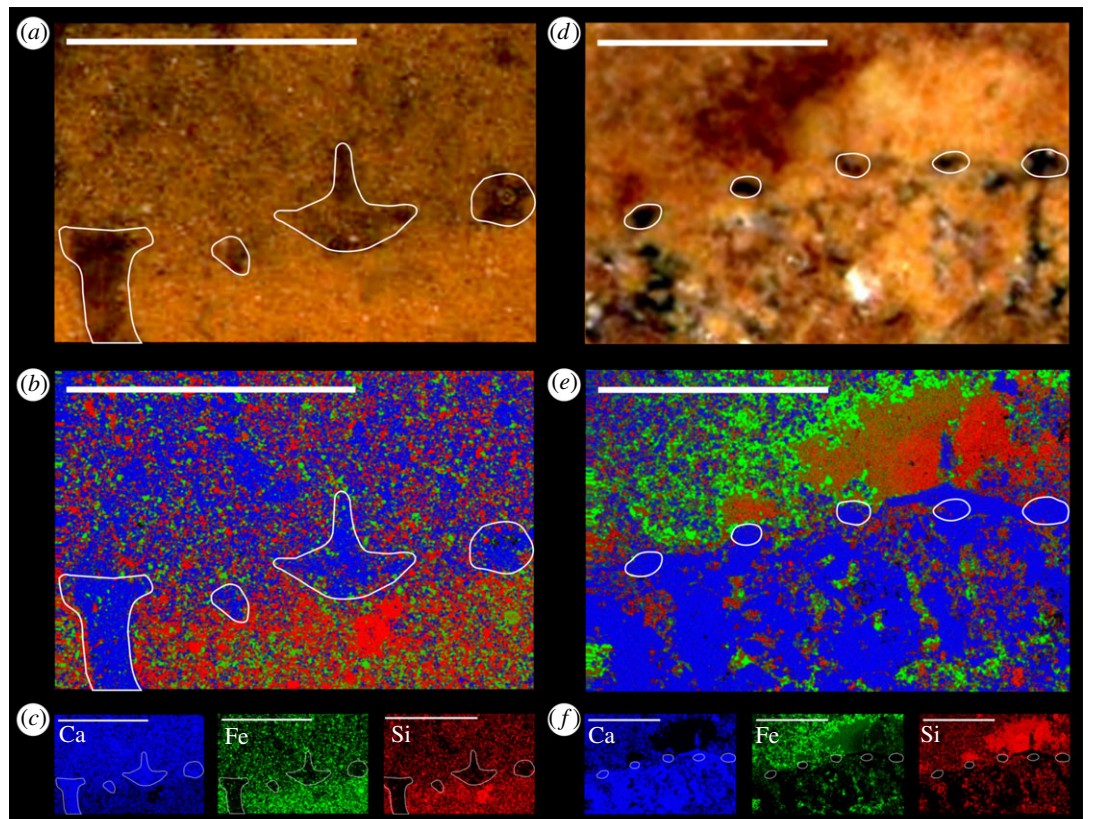

**Figure 6.** Elemental maps of *Carduispongia pedicula* gen. et sp. nov. spicules. (*a–c*) OUMNH C.36062, section through dermal surface intersecting four spicules, the left-most of which includes a prostalial ray, and another of which (third from left) includes a partial gastral ray. Interior of specimen is up. (*d–f*) OUMNH C.36061, section through dermal surface intersecting the dermal rays of five spicules. Interior of specimen is down. Abundances of Ca (present in calcite and ankerite, but not silica), Si (in silica but not ankerite or calcite), and Fe (in ankerite but not calcite or silica) are shown. (*a,d*) Details of photograph of region with approximate spicule margins indicated in white, (*b,e*) combined elemental maps of Ca (blue), Fe (green) and Si (red), with spicule positions marked. (*c,f*) Maps of these three elements in isolation. Single-pixel resolution 2.14 µm (*a–c*), and 2.81 µm (*d–f*). Scale bars 1 mm.

void-fill calcite occurred together. This is common in other calcitic Herefordshire fossils, for example, ostracods and brachiopods [26,27], where soft tissue and biomineralized calcite structures have merged as part of the preservation process. Siliceous spicules are also characterized by an axial canal, which is lacking in *Carduispongia* spicules. Bimineralic spicules comprising a silica core surrounded by calcite are known to occur in two Cambrian sponges, the heteractinid *Eiffelia* [13,28,29] and the protomonaxonid *Lenica* [13], but *Carduispongia* spicules lack any axial structures. Although taphonomic loss of a putative siliceous core cannot be ruled out entirely, the preservation of fine siliceous structures in other Herefordshire taxa, e.g. at a resolution of 10 µm or better in a radiolarian (figure 6*e*), suggests that this is unlikely. Sponge spicules are only known to biomineralize in silica or calcite. Non-biomineralized spicules (as opposed to relatively amorphous spongin fibres and spiculoids in *Darwinella* [30]) are unknown in extant or extinct sponges. Carbonaceous preservation of hexactine spicules has been described from lower Cambrian strata [31], but this represents organic sheaths surrounding an originally mineralized spicule [31]. Hypothetical 'soft' spicules would presumably lack the apparent rigidity of the prostalia observed here. We, therefore, conclude that the spicules of *C. pedicula* were biomineralized exclusively in calcite.

Hexactine spicules are restricted to the (siliceous) Hexactinellida among extant sponges, but hexactins are also known from many Palaeozoic groups that are likely to belong outside Hexactinellida, Silicea, or even crown-group Porifera [16]. Where the original composition is known, however, almost all hexactins are siliceous or bimineralic. Isolated Palaeozoic hexactine spicules preserved in calcite have been reported under the generic name *Calcihexactina* (e.g. [32–34]), but their original mineralogy is unclear [16] and some (e.g. [32]) possess a true axial canal that implies secondary calcite-replacement of an originally siliceous/bimineralic composition. Some '*Calcihexactina*' occurrences may represent primary calcitic

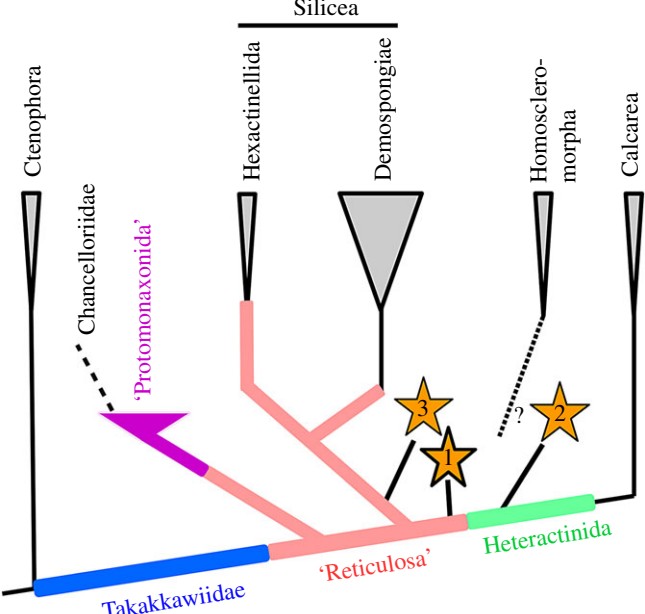

**Figure 7.** Possible phylogenetic positions for *Carduispongia* gen. nov., based on the evolutionary framework of Botting & Muir [16]: 1, preferred placement, requiring loss of siliceous core, independent of Calcarea; 2, derivation from heteractinids, involving loss of hetereractine spicules; 3, derivation from stem group of Silicea, involving switch from silica to calcite, and development of calcarean-like soft tissues.

hexactins [16]. Spicules in the reticulosan *Protospongia* may also be calcitic [35], although examples from the Cambrian of Idaho preserve axial structures that suggest a bimineralic composition ([36] fig. 5.1). *Carduispongia* provides the first definitive evidence of calcareous hexactine spicules.

The phylogenetic model proposed by Botting & Muir [16] derives crown-group Porifera from a thin-walled sponge with bimineralic hexactine (and perhaps other) spicules. Calcite was subsequently lost in Silicea, silica in Calcarea, and hexactins were retained only in Hexactinellida. The most parsimonious position for *Carduispongia* (figure 7, position 1) is as a stem-group calcarean (± Homoscleromorpha). The absence of heteractinid spicules might suggest a position low in the stem group, below Heteractinida. The early heteractinid *Eiffelia globosa*, however, possessed bimineralic spicules and hexactins as well as heteractins [13,16] requiring the loss of the siliceous core in *Carduispongia* to be independent of that in heteractinids. Derivation from an *Eiffelia*-like ancestor (figure 7, position 2) would require the loss of heteractins as well as the siliceous core. Alternatively, *Carduispongia* might represent a reticulosan silicean which evolved a calcite spicule morphology independently of this transition in the stem lineage of Calcarea (figure 7, position 3) although we consider this scenario to be less plausible.

The quincuncial spicule arrangement (rays joining or overlapping at tips) of *C. pedicula* is similar to that of the reticulosans *Diagoniella* [37] and *Protospongia* (e.g. [38]), rather than that of *Cyathophycus* [39] in which spicule centres are found in every corner of the grid. Other thin-walled and early reticulosans (e.g. *Heminectere* [39]) had a quincuncial spicule-arrangement, and the strand-based *Cyathophycus* condition is a derived feature within the silicean stem lineage (*sensu* [16]). The arrangement in *C. pedicula* is therefore suggestive of a basal position, rather than a derived silicean one. In the Heteractinida, skeletal regularity was also much less obvious, although relict quadruled (quincuncial) organization was still present in *Eiffelia* [13]. The extremely regular grid organization in *C. pedicula*, therefore, suggests a close link to the pre-heteractinid protospongioids, rather than loss of heteractins from later stem-calcareans.

The prostalia are a distinctive feature of the new sponge, and superficially resemble those of the Asthenospongiidae, especially the Ordovician *Acutipuerilis* [40] from central Wales. However, the skeletal organization of that family is relatively far removed from the simple quincuncial array of *C. pedicula*, being disordered in all known species, and with distinct basal attachment spicules present in *Acutipuerilis*; a close relationship to the Asthenospongiidae is thus difficult to support on skeletal evidence. Compositional evidence is also ambiguous, as the original mineralogy of asthenospongiids is unknown. Spicules in *Acutipuerilis* are preserved as iron oxides after pyrite replacement; this is a common form of preservation of siliceous spicules in mudstone but calcareous fossils such as trilobites co-occuring with *Acutipuerilis* are preserved in the same way (JP Botting, LA Muir personal observation, 2019). Few other asthenospongiids are known, and mineralogical determination is equally problematic in all cases.

*Carduispongia pedicula* is unique among Palaeozoic sponges in yielding detailed information on soft-tissue organization. The preserved choanocyte chambers are substantially larger than silicean chambers, but fall within the known size range of such structures in extant calcareans (e.g. [41]), and their arrangement and frequent connections conform closely with choanocyte chambers in some living sponges. The aquiferous system is either sylleibid or leuconoid, and the chamber disposition closely recalls that of the sylleibid calcarean *Grantiopsis cylindrica* ([41], figure 1*c*) although the presence or absence of enlarged exhalent chambers cannot be confirmed. The arrangement is also compatible with a leuconid interpretation, but typical leuconid tissue architecture involves a more complex organization of small chambers embedded within soft tissues, connected by distinct canals that are not evident in *Carduispongia* (e.g. [42]). The Homoscleromorpha (probable sister-group of Calcarea [5]) also includes representatives with very similar open sylleibid to leuconoid choanosomal architecture composed of similarly large, interconnected chambers, such as in *Plakina* [43].

Both traditional morphological/ontogenetic evidence (e.g. [44–46]) and molecular phylogenies [41,47] indicate that the primitive state of the crown group of Calcarea was asconid. Botting & Muir ([16], table 2) also inferred that asconid/syconid states were likely to have been typical of the stem groups of Porifera, Silicea and Calcarea. The discovery in the Herefordshire Lagerstätte of a relatively complex aquiferous system in a 'reticulosan', regardless of its precise phylogenetic position, casts doubt on this hypothesis. A sylleibid/leuconoid grade could have evolved convergently (aquiferous system-grade is characterized by homoplasy in Calcarea at least [41]), but if *Carduispongia* were typical of reticulosans, and particularly of the calcarean stem-group, this would imply either that our understanding of the phylogeny of Calcarea is incorrect, or that a sycon-grade architecture at the base of crown Calcarea represents a reversal from an earlier more complex system instead of a plesiomorphy. However, the hypertrophied prostalia in *Carduispongia* are associated with hypertrophied gastral rays, which support the soft tissues. These rays may have allowed the development of relatively complex soft tissues, which later calcareans with small spicules and rays might not have maintained. The array of large chambers in *Carduispongia* may not have been more efficient than the thinner-walled chambers in the simplest modern calcareans.

# 5. Conclusion

Unusual character combinations in an individual taxon should not be used to underpin models of phylogeny, as homoplasy and character-state reversals cannot be ruled out in a single case. Nonetheless, each new character combination contributes to developing the overall framework. The new evidence from the new Herefordshire sponge supports the hypothesis that hexactine spicules are plesiomorphic for Porifera [16], by confirming that they occur in the absence of siliceous biomineralization. By contrast, the relatively complex aquiferous system in *Carduispongia* is not predicted by existing phylogenetic models, although it is easily derived from a simple ascon/sycon architecture similar to that of modern thin-walled calcareans or homoscleromorphs. The combination of features in *Carduispongia* strongly supports a skeletal continuum between primitive calcareans and hexactinellid siliceans, indicating that the last common ancestor of Porifera was a spiculate, solitary, vasiform animal with a thin body wall, perhaps with unexpectedly complex soft-tissue organization. Testing this hypothesis will require the study of more species of well-preserved stem-group Porifera, Calcarea and Silicea, such as those awaiting documentation in the rich sponge-fauna of the Herefordshire Lagerstätte.

Data accessibility. Datasets and the final 3-D model in VAXML/STL file format are available from the Dryad Digital Repository: https://doi.org/10.5061/dryad.8mv00jv [48].

Authors' contributions. D.E.G.B., D.J.S., D.J.S. and M.D.S. designed the research and conducted fieldwork. P.G. and A.K. performed synchrotron analysis. A.N. conducted reconstructions, descriptions and an initial draft of the manuscript. A.N., M.D.S., J.P.B. and L.A.M. analysed and interpreted the data. J.P.B. drew figures 2 and 7. All authors provided scientific and editorial input to the paper.

Competing interests. We have no competing interests.

Funding. This work was supported by the Yale Peabody Museum of Natural History Invertebrate Paleontology Division, the Natural Environment Research Council (grant no. NF/F0108037/1), the Leverhulme Trust (grant no. EM-2014-068) and English Nature.

Acknowledgements. We thank Carolyn Lewis for technical assistance and David Edwards and the late Roy Fenn for general assistance associated with fieldwork. We acknowledge SOLEIL synchrotron for provision of beamtime under project no. 20151314 and Serge X. Cohen (IPANEMA) for assistance during the experiment. We thank Marcelo Carrera and an anonymous reviewer for further helpful input.

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
