## [Reviewer comments · Royal Society Open Science]

Review History

RSOS-190911.R0 (Original submission)

Review form: Reviewer 1

Is the manuscript scientifically sound in its present form?

Yes

Are the interpretations and conclusions justified by the results?

Yes

Is the language acceptable?

Yes

Is it clear how to access all supporting data?

Yes

Do you have any ethical concerns with this paper?

No

Have you any concerns about statistical analyses in this paper?

No

Recommendation?

Accept with minor revision (please list in comments)

Comments to the Author(s)

This excellent and well-written manuscript provides a thorough description of an important new sponge with remarkable preservational fidelity. The novel combination of characteristics observed is skilfully applied to the emerging picture of sponge origins. The phylogenetic interpretations proposed are well defended, precise and clear; ambiguities are carefully considered, and the conclusions are measured and well justified. I have no hesitation in recommending the manuscript for publication.

I can see little in the manuscript that might be improved, but have a couple of minor suggestions.

The mixture of form-taxonomy with formal taxonomy makes me a little uncomfortable. If 'Reticulosa' may be polyphyletic, is it necessary to assign the genus to an Order at all, rather than leaving it in open nomenclature? This said, the paraphyletic 'Reticulosa' envisaged in Fig. 7 clearly has some taxonomic value.

P3L1. Suggest removing 'both' to avoid ambiguity (were both spicules preserved, as well as soft tissue?)

P4L32. 'stem-group' requires a subject; all fossils are by definition in *some* stem group. Suggest removing 'stem-group' or specifying 'fossils from the stem-group of X'.

P9L26. Is it worth specifying the evidence that this is calcite, rather than any other polymorph of CaCO_3 ? Presumably the presence of Ca alone does not discount ankerite? I wonder whether there might be value in mentioning the number of specimens in which mineralogy has been determined, to assuage possible concerns that the absence of ankerite or primary silica may reflect a chance variation in taphonomic regime.

P10L30. The authors may wish to consider discussing the carbonaceous hexactine spicules described by Harvey (2010, *Biology Letters*) as examples of semi-rigid non-mineralized (demineralized?) spicules.

Fig 1. The density of panels is very high, making it difficult to benefit from the stereo pairs; could these be given a little more room, such that they can be viewed more comfortably?

Note that abbreviations are out of order in the caption (cc before co).

The captions to Figs 5 & 6 seem to have been swapped in the PDF proof.

Review form: Reviewer 2 (Marcelo Carrera)**Is the manuscript scientifically sound in its present form?**

Yes

Are the interpretations and conclusions justified by the results?

Yes

Is the language acceptable?

Yes

Is it clear how to access all supporting data?

Yes

Do you have any ethical concerns with this paper?

No

Have you any concerns about statistical analyses in this paper?

No

Recommendation?

Accept with minor revision (please list in comments)

Comments to the Author(s)

Dear Editors of the Royal Society

The paper: "Three-dimensionally preserved soft-tissues and calcareous hexactins in a Silurian sponge: implications for early sponge evolution" is an important contribution to the knowledge of the early evolutionary history of the Phylum Porifera.

The authors present compelling evidence for an original biomineralization of hexactinellid spicules "calci-hexactines" that confirm the new vision of the early evolution of sponges with diffuse limits among Classes, in contrast to the well differentiated taxonomy among extant sponges. Besides, they show with a 3D technic the organic preservation of the wall morphology including internal chambers, which is extremely rare among fossil sponge preservation.

I think the paper is of broad interest among paleontologists and neontologist.

Minor suggestions:

A location map and column could improve to visualize the geology of the area for open scope readers.

An additional paragraph showing previous thoughts about the explanation for taphonomic replacement of silica by calcite in sponges could help also the non-familiarized readers.

Sincerely,

Marcelo G. Carrera

Decision letter (RSOS-190911.R0)

19-Jun-2019

Dear Dr Sutton

On behalf of the Editors, I am pleased to inform you that your Manuscript RSOS-190911 entitled "Three-dimensionally preserved soft-tissues and calcareous hexactins in a Silurian sponge: implications for early sponge evolution." has been accepted for publication in Royal Society Open

Science subject to minor revision in accordance with the referee suggestions. Please find the referees' comments at the end of this email.

The reviewers and handling editors have recommended publication, but also suggest some minor revisions to your manuscript. Therefore, I invite you to respond to the comments and revise your manuscript.

- Ethics statement

- Data accessibility

If you wish to submit your supporting data or code to Dryad (<http://datadryad.org/>), or modify your current submission to dryad, please use the following link:
<http://datadryad.org/submit?journalID=RSOS&manu=RSOS-190911>

- Competing interests

- Authors' contributions

- Acknowledgements

- Funding statement

Because the schedule for publication is very tight, it is a condition of publication that you submit the revised version of your manuscript before 28-Jun-2019. Please note that the revision deadline will expire at 00.00am on this date. If you do not think you will be able to meet this date please let me know immediately.

Supplementary files will be published alongside the paper on the journal website and posted on the online figshare repository (<https://rs.figshare.com/>). The heading and legend provided for each supplementary file during the submission process will be used to create the figshare page,

so please ensure these are accurate and informative so that your files can be found in searches. Files on figshare will be made available approximately one week before the accompanying article so that the supplementary material can be attributed a unique DOI.

on behalf of Kevin Padian (Subject Editor)
openscience@royalsociety.org

Reviewer comments to Author:
Reviewer: 1

Comments to the Author(s)

This excellent and well-written manuscript provides a thorough description of an important new sponge with remarkable preservational fidelity. The novel combination of characteristics observed is skilfully applied to the emerging picture of sponge origins. The phylogenetic interpretations proposed are well defended, precise and clear; ambiguities are carefully considered, and the conclusions are measured and well justified. I have no hesitation in recommending the manuscript for publication.

I can see little in the manuscript that might be improved, but have a couple of minor suggestions.

The mixture of form-taxonomy with formal taxonomy makes me a little uncomfortable. If 'Reticulosa' may be polyphyletic, is it necessary to assign the genus to an Order at all, rather than leaving it in open nomenclature? This said, the paraphyletic 'Reticulosa' envisaged in Fig. 7 clearly has some taxonomic value.

P3L1. Suggest removing 'both' to avoid ambiguity (were both spicules preserved, as well as soft tissue?)

P4L32. 'stem-group' requires a subject; all fossils are by definition in *some* stem group. Suggest removing 'stem-group' or specifying 'fossils from the stem-group of X'.

P9L26. Is it worth specifying the evidence that this is calcite, rather than any other polymorph of CaCO_3 ? Presumably the presence of Ca alone does not discount ankerite? I wonder whether there might be value in mentioning the number of specimens in which mineralogy has been determined, to assuage possible concerns that the absence of ankerite or primary silica may reflect a chance variation in taphonomic regime.

P10L30. The authors may wish to consider discussing the carbonaceous hexactine spicules described by Harvey (2010, *Biology Letters*) as examples of semi-rigid non-mineralized (demineralized?) spicules.

Fig 1. The density of panels is very high, making it difficult to benefit from the stereo pairs; could these be given a little more room, such that they can be viewed more comfortably?

Note that abbreviations are out of order in the caption (cc before co).

The captions to Figs 5 & 6 seem to have been swapped in the PDF proof.

Reviewer: 2

Comments to the Author(s)

Dear Editors of the Royal Society

The paper: "Three-dimensionally preserved soft-tissues and calcareous hexactins in a Silurian sponge: implications for early sponge evolution" is an important contribution to the knowledge of the early evolutionary history of the Phylum Porifera.

The authors present compelling evidence for an original biomineralization of hexactinellid spicules "calci-hexactines" that confirm the new vision of the early evolution of sponges with diffuse limits among Classes, in contrast to the well differentiated taxonomy among extant sponges. Besides, they show with a 3D technic the organic preservation of the wall morphology including internal chambers, which is extremely rare among fossil sponge preservation.

I think the paper is of broad interest among paleontologists and neontologist.

Minor suggestions:

A location map and column could improve to visualize the geology of the area for open scope readers.

An additional paragraph showing previous thoughts about the explanation for taphonomic replacement of silica by calcite in sponges could help also the non-familiarized readers.

Sincerely,

Marcelo G. Carrera

Author's Response to Decision Letter for (RSOS-190911.R0)

See Appendix A.

Decision letter (RSOS-190911.R1)

09-Jul-2019

Dear Dr Sutton,

I am pleased to inform you that your manuscript entitled "Three-dimensionally preserved soft-tissues and calcareous hexactins in a Silurian sponge: implications for early sponge evolution." is now accepted for publication in Royal Society Open Science.

on behalf of Kevin Padian (Subject Editor)
openscience@royalsociety.org

Follow Royal Society Publishing on Twitter: [@RSocPublishing](https://twitter.com/RSocPublishing)

Appendix A

Dear Kevin,

We thank the two reviewers for their enthusiastic endorsement. Responses to their particular comments and suggestions are presented below (in **bold**). All changes to the manuscript have been tracked; some changes are minor editorial tweaks of wording rather than responses to reviewers, so we have identified changes made in response to review by highlighting in yellow. As requested, we also provide a clean version of the manuscript with all tracking and highlighting removed.

Please note that while the instructions we receive ask for the supplementary information to be in final form, ours still includes highlighted slots for the final published DOI and year, as of course we cannot provide these at present.

Best wishes,

Mark Sutton

Reviewer: 1

Comments to the Author(s)

This excellent and well-written manuscript provides a thorough description of an important new sponge with remarkable preservational fidelity. The novel combination of characteristics observed is skilfully applied to the emerging picture of sponge origins. The phylogenetic interpretations proposed are well defended, precise and clear; ambiguities are carefully considered, and the conclusions are measured and well justified. I have no hesitation in recommending the manuscript for publication.

I can see little in the manuscript that might be improved, but have a couple of minor suggestions.

The mixture of form-taxonomy with formal taxonomy makes me a little uncomfortable. If 'Reticulosa' may be polyphyletic, is it necessary to assign the genus to an Order at all, rather than leaving it in open nomenclature? This said, the paraphyletic 'Reticulosa' envisaged in Fig. 7 clearly has some taxonomic value.

As the reviewer notes, the paraphyletic 'Reticulosa' retains some practical value as a label, and hence we prefer to retain it as-is in our manuscript. Workers on fossil sponges need to have a concept of what a reticulosan is; we feel that removing it from our taxonomy would degrade the utility of our work. While of course we agree that paraphyletic taxa are not desirable, we prefer to indicate this through open taxonomy and discussion, rather than simply to excise the name.

P3L1. Suggest removing 'both' to avoid ambiguity (were both spicules preserved, as well as soft tissue?)
Fixed ('both' removed as suggested)

P4L32. 'stem-group' requires a subject; all fossils are by definition in *some* stem group. Suggest removing 'stem-group' or specifying 'fossils from the stem-group of X'.
Fixed ('stem-group' removed to resolve the tautology)

P9L26. Is it worth specifying the evidence that this is calcite, rather than any other polymorph of CaCO₃? Presumably the presence of Ca alone does not discount ankerite? I wonder whether there might be value in mentioning the number of specimens in which mineralogy has been determined, to assuage possible concerns that the absence of ankerite or primary silica may reflect a chance variation in taphonomic regime.

The text does state that calcite is evident optically. Without complex crystallographic work there is no simple way to differentiate calcite from aragonite, but these two polymorphs are relatively

straightforward to distinguish optically. Aragonite is in any case unstable geologically, and would be very unexpected in a Palaeozoic fossil. No other polymorphs of calcite occur geologically. As the reviewer points out, Ca alone does not discount ankerite, but the spicules are also depleted in iron, which should be present in ankerite. Elemental mapping was only performed on two specimens, but all specimens examined by eye show calcite-preservation of spicules (a specimen count is already provided). We have modified this paragraph to clarify these points.

P10L30. The authors may wish to consider discussing the carbonaceous hexactine spicules described by Harvey (2010, *Biology Letters*) as examples of semi-rigid non-mineralized (demineralized?) spicules. **We thank the reviewer for suggesting this paper, and have added a mention. These spicules are interpreted by the original author as organic sheaths encasing a mineralised (but not fossilised) spicule, so do not change the direction of our discussion – but the paper should certainly be cited as we have done.**

Fig 1. The density of panels is very high, making it difficult to benefit from the stereo pairs; could these be given a little more room, such that they can be viewed more comfortably?

There is little scope to give any more room without shrinking the images; we have published many similar plates before (in Royal Society journals), including others denser than this, and the stereo-pair separation is on a par with that we have used previously. We have re-checked to confirm that the stereo-pairs work perfectly well; any resizing would risk impairing the stereo effect.

Note that abbreviations are out of order in the caption (cc before co).

Thanks - fixed

The captions to Figs 5 & 6 seem to have been swapped in the PDF proof.

Thanks - fixed

Reviewer: 2

Comments to the Author(s)

The paper: “Three-dimensionally preserved soft-tissues and calcareous hexactins in a Silurian sponge: implications for early sponge evolution” is an important contribution to the knowledge of the early evolutionary history of the Phylum Porifera.

The authors present compelling evidence for an original biomineralization of hexactinellid spicules “calci-hexactines” that confirm the new vision of the early evolution of sponges with diffuse limits among Classes, in contrast to the well differentiated taxonomy among extant sponges. Besides, they show with a 3D technic the organic preservation of the wall morphology including internal chambers, which is extremely rare among fossil sponge preservation.

I think the paper is of broad interest among paleontologists and neontologist.

Minor suggestions:

A location map and column could improve to visualize the geology of the area for open scope readers.

We prefer not to provide these, having published many papers on elements of this fauna without them. This information belongs in a review article instead, and we have just submitted one such to the *Journal of the Geological Society*. We could add a citation to this at proof stage, assuming it is in press by then.

An additional paragraph showing previous thoughts about the explanation for taphonomic replacement of silica by calcite in sponges could help also the non-familiarized readers.

We are not claiming any replacement of silica by calcite in the paper. We do claim a replacement of silica by ankerite, but we already discuss this, providing references. We do also briefly mention that some ‘*Calcihexactina*’ occurrences may represent replacement of silica by calcite, but to the best of

our knowledge no taphonomic mechanism for this has been proposed, and speculation about this is beyond the scope of this paper.